# Association between the novel inflammatory marker white blood cell count-to-mean platelet volume ratio and metabolic syndrome: A cross-sectional study based on NHANES 2011–2020

Shuiying Huang[1], Yi Wang[2], Yu Wang[1]*, Ying Miao[3]*

1 Department of Cardiology, Luzhou People's Hospital, Luzhou, China, 2 Department of Pharmacy, Luzhou People's Hospital, Luzhou, China, 3 Department of Endocrinology and Metabolism, Affiliated Hospital of Southwest Medical University, Luzhou, China

* 317572779@qq.com (YW); 249675761@qq.com (YM)

## Abstract

### Background

Metabolic syndrome (MetS) is a chronic non-infective syndrome characterized by a set of vascular risk factors, including insulin resistance, hypertension, abdominal obesity, impaired glucose metabolism, and dyslipidaemia. This study aims to investigate the potential association between a novel inflammatory marker, the white blood cell count-to-mean platelet volume ratio (WMR), and MetS. By examining this association, we seek to provide data supporting the effective prevention of MetS through the improvement of inflammatory responses.

### Methods

We conducted a cross-sectional study using data from adult participants in the National Health and Nutrition Examination Survey (NHANES) from 2011 to 2020. Comprehensive data on complete blood count parameters and MetS were collected. MetS was defined according to the Adult Treatment Program III of the National Cholesterol Education Program. The formula for calculating WMR is: WMR = white blood cell count (1000 cells/μl)/ mean platelet volume (fL). Participants were stratified into four quartile groups (Q1 to Q4) based on their WMR levels, and chi-square tests along with rank-sum tests were utilized to assess differences in variables. Spearman correlation analysis was employed to evaluate the association between WMR and risk factors linked to MetS and other clinical indicators. Logistic regression analysis and subgroup analysis were conducted to investigate the independent interaction between WMR and MetS, and to further explore the association between WMR levels and the five specific components of MetS. Finally, receiver operating characteristic (ROC) curve analysis was performed to assess the predictive accuracy of WMR for MetS.

**Data availability statement:** The data underlying this study may be found in the following repository: https://doi.org/10.6084/m9.figshare.29557250.

**Funding:** The author(s) received no specific funding for this work.

**Competing interests:** NO authors have competing interests.

## Results

A total of 4917 participants were included in this study, comprising 2460 males and 2457 females. Among them, 1717 individuals (34.92%) were diagnosed with MetS. As quartile groups of WMR increased, the rates of MetS occurrence and its components, including Elevated FPG, Elevated TG, Elevated WC, and Low HDL-C, also increased. Spearman correlation analysis demonstrated a positive correlation between WMR and the insulin resistance index HOMA-IR. Logistic regression analysis, after adjusting for multiple confounders, revealed that each standard deviation increase in WMR was associated with a significant 3.185-fold increase in the odds of MetS prevalence (95% CI, 2.399–4.229; $P < 0.001$). In logistic regression analysis based on WMR quartiles (Q1 to Q4), the risks of MetS were 1.285 (95% CI, 1.045–1.582; $P < 0.001$), 1.586 (95% CI, 1.288–1.953; $P < 0.001$), and 2.548 (95% CI, 2.067–3.140; $P < 0.001$), respectively. After adjustment for multiple confounders, WMR levels were positively associated with Elevated FPG (OR = 2.126; 95% CI, 1.599–2.826; $P < 0.001$), Elevated TG (OR = 2.893; 95% CI, 2.095–3.995; $P < 0.001$), Elevated WC (OR = 2.678; 95% CI, 1.969–3.643; $P < 0.001$), and Low HDL-C (OR = 2.770; 95% CI, 2.049–3.744; $P < 0.001$). Subgroup analysis and interaction tests demonstrated that gender, age, race, education, smoking status, and physical activity modified the positive association between WMR and MetS (p for interaction < 0.05). Additionally, ROC curve analysis showed that the optimal cutoff value for WMR predicting MetS was 0.7974 (sensitivity: 58.4%; specificity: 59.9%; AUC: 0.621).

## Conclusions

Increasing WMR levels are significantly associated with the risk of MetS and its components: Elevated FPG, Elevated TG, Low HDL-C, and Elevated WC. This suggests that WMR could potentially serve as a valuable and reliable biomarker for MetS, highlighting the importance of closely monitoring patients with elevated WMR to improve prevention and mitigate the development of MetS. However, prospective cohort studies are warranted to confirm these associations and to further explore the causal relationships between WMR and the development of MetS.

## Introduction

With many ancient infectious diseases successfully controlled globally, non-communicable diseases have become the leading causes of morbidity and mortality in both developed and developing countries. Metabolic syndrome (MetS) is a truly global issue. It is not a single disease but a cluster of cardiovascular disease risk factors, with slightly different definitions across various organizations [1]. According to statistics, the prevalence of MetS in the United States increased from 25.29% in 1988 to 34.7% in 2016 [2]. Therefore, MetS has emerged as a serious public health concern, supported by mounting evidence linking it to increased risks of coronary

heart disease, cardiovascular diseases, type 2 diabetes, stroke, and overall mortality [1]. Given the substantial threat that MetS poses to human health, early detection and intervention have become pivotal topics of interest among scholars in relevant fields.

MetS also known as insulin resistance syndrome [3], is primarily driven by insulin resistance as its pathophysiological mechanism. A pro-inflammatory state is widely acknowledged as a component of MetS [4]. Chronic low-grade inflammation and activation of the immune system play roles in the pathogenesis of insulin resistance associated with obesity. Adipose tissue, liver, muscle, and pancreas are inflammatory sites in obesity. Infiltration of macrophages and other immune cells into these tissues is observed, correlating with a shift from an anti-inflammatory to a pro-inflammatory state among cell populations. These cells are pivotal in generating pro-inflammatory cytokines, which disrupt insulin signaling in peripheral tissues through autocrine and paracrine mechanisms [5], thereby promoting the onset of MetS.

White blood cells (WBCs) play a critical role in systemic inflammation. A study demonstrated that patients with MetS exhibit elevated WBC counts and changes in WBC subtypes including monocytes, neutrophils, and lymphocytes, which are positively correlated with body mass index, body fat percentage, and insulin resistance [6]. Mean platelet volume (MPV) reflects the size of platelets and offers insights into platelet function and activation [7]. Under inflammatory conditions, MPV is also linked to an increase in the proportion of larger platelets [8]. White blood cell count-to-mean platelet volume ratio (WMR), a composite marker comprising WBC count and MPV, serves as a novel inflammatory indicator [9]. To date, the association between WMR and MetS has not been established, and its underlying mechanisms remain unclear. Therefore, this cross-sectional study aims to explore the association between WMR and the risk of developing MetS, aiming to identify a straightforward marker for assessing MetS risk and facilitating early intervention to mitigate its occurrence.

## Methods

### Study subjects

Fig 1 illustrates the process of selecting study subjects from the National Health and Nutrition Examination Survey (NHANES). We excluded 3,568 participants due to missing age data, 17,800 participants who were under 18 years old, 279 pregnant women, 19,147 participants with incomplete MetS data, and 10 participants with incomplete blood routine examination data. Ultimately, the study included 4,917 participants.

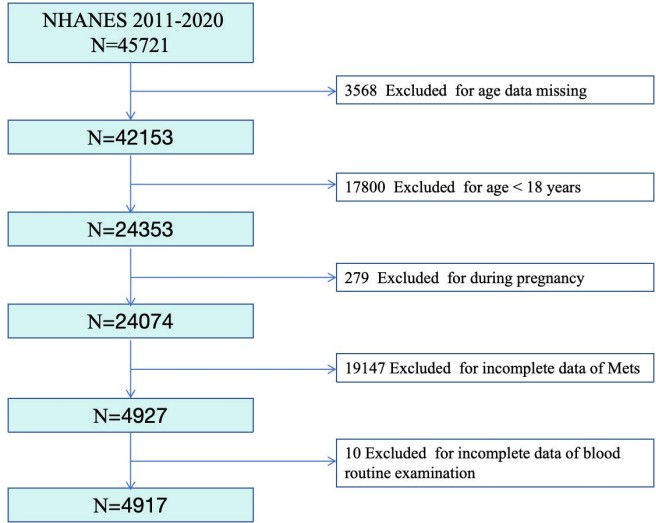

**Fig 1. Flowchart for the Selection of the Analyzed Study Sample From the NHANES.**

## Data collection

Interviewer-administered questionnaires were used to collect demographic information (age, gender, race) and educational attainment. Lifestyle factors, including cigarette use, alcohol consumption, and physical activity, were also assessed. Cigarette use was classified as never, former, or current. Never smokers were defined as individuals who had smoked fewer than 100 cigarettes in their lifetime. Former smokers were those who had smoked at least 100 cigarettes in their lifetime but were not currently smoking. Current smokers were defined as individuals who had smoked at least 100 cigarettes in their lifetime and reported current cigarette use. Participants were asked to self-report the frequency and duration of moderate and vigorous physical activities related to work, transportation, and leisure. From these responses, we derived a dichotomous variable indicating whether participants met the 2008 U.S. national physical activity guidelines. These guidelines recommend achieving at least 150 minutes of moderate activity, 75 minutes of vigorous activity per week, or an equivalent combination [10]. To reduce confounding and improve the accuracy of white blood cell (WBC) measurements, we applied strict exclusion criteria. Participants were excluded if they had conditions or exposures known to significantly influence WBC levels, including: Diagnosed chronic inflammatory diseases (e.g., rheumatoid arthritis, systemic lupus erythematosus), active malignancy or history of cancer treatment within the past year, acute infections at the time of examination, use of immunosuppressive agents, glucocorticoids, cytotoxic chemotherapy, or other medications that could alter immune cell counts, pregnancy, due to known changes in hematologic parameters. These exclusions were based on medical history, laboratory findings, medication use, and questionnaire data collected by NHANES.

## Ethics statement

NHANES is conducted by the Centers for Disease Control and Prevention (CDC) and the National Center for Health Statistics (NCHS). The NCHS Research Ethics Review Committee reviewed and approved the NHANES study protocol. All participants signed written informed consent.

## Definition

(1). WMR is calculated using the formula: WMR = White Blood Cell Count (1000 cells/µl)/ Mean Platelet Volume (fL) [9].

(2). Insulin resistance was assessed using the HOMA method with the equation: HOMA-IR = [Fasting insulin (µU/mL) × Fasting glucose (mmol/L)]/ 22.5 [11].

(3). Body mass index (BMI) was calculated using weight in kilograms divided by height in meters squared [12].

(4). Waist-to-height ratio (WtHR) was calculated as waist circumference (cm) divided by height (cm) [13].

## MetS definition

MetS is diagnosed using the criteria from the Adult Treatment Program III of the National Cholesterol Education Program [14]. To meet the MetS diagnosis, an individual must exhibit any three of the following five criteria: (1) TG ≥ 150 mg/dL; (2) HDL-C < 40 mg/dL in men and < 50 mg/dL in women; (3) FPG ≥ 100 mg/dL; (4) waist circumference (WC) > 102 cm in men and > 88 cm in women; (5) systolic blood pressure (SBP) of ≥ 130 mmHg and/or a diastolic blood pressure (DBP) of ≥ 85 mmHg. Fasting blood samples were collected in the morning after a 9-hour fast, and BP was measured three times by a physician to determine the average value.

## Statistical analysis

The characteristics of study participants were presented as either mean (standard deviation) or median (interquartile range), depending on the distribution of continuous variables. Categorical variables were expressed as count (proportion).

For comparisons of continuous variables, Student's t-test, Mann-Whitney U test, Kruskal-Wallis H test, or one-way ANOVA were utilized based on the normality of the data. Chi-square tests were employed for comparing categorical variables between groups.

Logistic regression models were used to assess the association between WMR and MetS and its components, presenting results as odds ratios (OR) with corresponding 95% confidence intervals (CI). Subgroup analyses investigating the association between WMR and MetS were conducted using stratified factors: sex, age, race, education, smoking status, and physical activity. These stratification variables were also considered as pre-specified possible effect modifiers, and interactions between WMR and these factors were tested using interaction terms.

The predictive validity of WMR for MetS presence was evaluated using receiver operating characteristic (ROC) curves and area under the curve (AUC) across all subjects. Statistical significance was determined using two-tailed p-values with a threshold set at $p < 0.05$. Statistical analyses were performed using SPSS (version 26.0), and Forest plots were generated using GraphPad Prism (version 9.0.0).

## Results

### Clinical characteristics of the participants

The clinical characteristics of the subjects are shown in Table 1. A total of 4917 subjects were included in this study, with 2460 males and 2457 females, with an average age of 43 years. Among the participants, 34.92% were diagnosed with MetS, 51.70% with elevated FPG, 44.21% with elevated BP, 21.90% with elevated TG, 31.32% with low HDL-C, and 53.30% with elevated WC.

Participants were divided into four groups (Q1-Q4) based on WMR quartiles. A comparison of clinical and laboratory characteristics among these groups revealed that from Q1 to Q4, the proportion of current smokers and participants who drink more than three glasses per day gradually increased, with statistically significant differences ($P < 0.05$). BMI, WHtR, WBC, red blood cell count (RBC), hemoglobin (Hb), platelet count (PLT), and homeostasis model assessment of insulin resistance (HOMA-IR) all gradually increased, with statistically significant differences ($P < 0.05$). Conversely, the proportion of physically active individuals gradually decreased from the lowest to the highest quartile groups. The proportions of diabetes, hyperlipidemia, hyperuricemia, MetS, elevated FPG, elevated TG, low HDL-C, and elevated WC all gradually increased, with statistically significant differences ($P < 0.05$).

### Association between WMR and clinical/laboratory characteristics

Spearman correlation analysis, shown in Table 2, indicates that WMR is positively correlated with WC, BMI, WHtR, FPG, postprandial 2-hour plasma glucose (2hPG), hemoglobin A1c (HbA1c), HOMA-IR, TG, WBC, RBC, Hb, PLT, aspartate aminotransferase (AST), gamma-glutamyl transferase (GGT), uric acid, and MetS, with statistically significant differences ($P < 0.05$). Conversely, it is negatively correlated with alanine aminotransferase (ALT) and HDL-C, also with statistically significant differences ($P < 0.05$).

### Univariate analysis of determinants of MetS in study subjects

Table 3 shows the associations of WMR and other variables with the risk of MetS presence. Univariate analysis revealed significant associations between MetS and Age, Gender, Race, Education, Smoking status, Alcohol consumption, Physical activity, PIR, RBC, AST, GGT, TC, LDL-C, uric acid, and WMR.

### Association of WMR with MetS and its components

The results of logistic regression analysis examining the association between WMR and MetS are presented in Table 4. The relationship was found to be significant across different models: in the unadjusted crude model (model 1) (odds ratio

**Table 1. Baseline Characteristics of the participants based on WMR Categories.**

| Variable | Total | Q1(n=1227) [0.23-0.6395] | Q2(n=1233) (0.6395~0.7778] | Q3(n=1225) (0.7778~0.9487] | Q4(n=1232) (0.9487~6.75] | P value |
|---|---|---|---|---|---|---|
| Age, years | 43.00(30.00,61.00) | 45.00(30.00,61.00) | 44.00(29.00,62.00) | 44.00(29.50,62.00) | 42.00(30.00,59.00) | 0.676 |
| Gender % | | | | | | 0.572 |
| Male | 2460 | 604(24.60%) | 632(25.70%) | 622(25.30%) | 602(24.50%) | |
| Female | 2457 | 623(25.40%) | 601(24.50%) | 603(24.50%) | 630(25.60%) | |
| Race % | | | | | | <0.001 |
| Mexican American | 696 | 149(21.40%) | 168(24.10%) | 178(25.60%) | 201(28.90%) | |
| Other Hispanc | 582 | 121(20.80%) | 146(25.10%) | 176(30.20%) | 139(23.90%) | |
| Non-Hispanic White | 1779 | 369(20.70%) | 438(24.60%) | 478(26.90%) | 494(27.80%) | |
| Non-Hispanic Black | 1042 | 390(37.40%) | 254(24.40%) | 189(18.10%) | 209(20.10%) | |
| Other race | 818 | 198(24.20%) | 227(27.80%) | 204(24.90%) | 189(23.10%) | |
| Education % | | | | | | <0.001 |
| Less than high school | 1009 | 217(21.50%) | 228(22.60%) | 262(26.00%) | 302(29.90%) | |
| High school diploma | 1033 | 268(25.90%) | 246(23.80%) | 247(23.90%) | 272(26.30%) | |
| More than high school | 2568 | 660(25.70%) | 678(26.40%) | 644(25.10%) | 586(22.80%) | |
| Smoking status % | | | | | | <0.001 |
| Never smoker | 2890 | 830(28.70%) | 777(26.90%) | 704(24.40%) | 579(20.00%) | |
| Former smoker | 1072 | 263(24.50%) | 283(26.40%) | 276(25.70%) | 250(23.30%) | |
| Current smoker | 955 | 134(14.00%) | 173(18.10%) | 245(25.70%) | 403(42.20%) | |
| Alcohol consumption % | | | | | | <0.001 |
| 0-1 cup/day | 2801 | 726(25.90%) | 707(25.20%) | 712(25.40%) | 656(23.40%) | |
| 2-3 cups/day | 1362 | 361(26.50%) | 348(25.60%) | 293(21.50%) | 360(26.40%) | |
| >3 cups/day | 754 | 140(18.60%) | 178(23.60%) | 220(29.20%) | 216(28.60%) | |
| BMI | 27.80(24.00,32.60) | 26.50(23.10,30.50) | 27.60(23.60,31.90) | 28.00(24.30,32.50) | 29.55(25.20,35.60) | <0.001 |
| WHtR | 0.58(0.52,0.65) | 0.55(0.49,0.62) | 0.58(0.51,0.64) | 0.59(0.52,0.66) | 0.61(0.54,0.69) | <0.001 |
| Laboratory tests | | | | | | |
| WBC, ×10$^9$/L | 6.50(5.40,7.80) | 4.70(4.20,5.30) | 6.00(5.60,6.50) | 7.00(6.50,7.60) | 9.00(8.10,10.10) | <0.001 |
| RBC, ×10$^{12}$/L | 4.72(4.39,5.05) | 4.62(4.31,4.97) | 4.71(4.40,5.07) | 4.74(4.40,5.06) | 4.77(4.48,5.12) | <0.001 |
| Hb, mg/dL | 14.10(13.10,15.20) | 13.90(13.00,14.90) | 14.10(13.10,15.10) | 14.20(13.20,15.20) | 14.30(13.20,15.40) | <0.001 |
| PLT, ×10$^9$/L | 230.00(195.00,272.00) | 199.00(173.00,232.00) | 220.00(190.00,254.00) | 238.00(205.00,273.00) | 271.50(232.25,314.00) | <0.001 |
| ALT, U/L | 22.00(19.00,27.00) | 22.00(19.00,28.00) | 22.00(19.00,27.00) | 22.00(18.00,27.00) | 21.00(18.00,26.00) | 0.184 |

*(Continued)*

**Table 1.** (Continued)

| Variable | Total | Q1(n=1227) [0.23-0.6395] | Q2(n=1233) (0.6395~0.7778] | Q3(n=1225) (0.7778~0.9487] | Q4(n=1232) (0.9487~6.75] | P value |
|---|---|---|---|---|---|---|
| AST, U/L | 20.00(15.00,28.00) | 20.00(15.00,27.00) | 20.00(15.00,28.00) | 20.00(15.00,28.00) | 21.00(15.00,29.00) | <0.001 |
| GGT, U/L | 19.00(14.00,29.00) | 18.00(13.00,27.00) | 18.00(13.00,28.00) | 20.00(14.00,29.00) | 21.00(15.00,33.00) | <0.001 |
| TC, mg/dL | 183.00(157.00,211.00) | 183.00(157.00,210.00) | 183.00(158.00,210.50) | 183.00(159.00,212.00) | 183.00(155.25,212.00) | 0.812 |
| LDL-C, mg/dL | 107.00(85.00,132.00) | 106.00(84.00,130.00) | 107.00(85.75,133.00) | 107.00(86.00,133.00) | 107.00(84.00,132.00) | 0.672 |
| HOMA-IR | 2.44(1.44,4.37) | 1.86(1.19,3.25) | 2.36(1.41,3.98) | 2.61(1.55,4.56) | 3.21(1.77,5.81) | <0.001 |
| PIR | 1.99(1.02,3.87) | 2.23(1.09,4.09) | 2.11(1.02,4.18) | 1.85(0.97,3.67) | 1.77(0.96,3.47) | <0.001 |
| Physically active % | | | | | | <0.001 |
| Yes | 1735 | 504(29.00%) | 493(28.40%) | 378(21.80%) | 360(20.70%) | |
| No | 3182 | 723(22.70%) | 740(23.30%) | 847(26.60%) | 872(27.40%) | |
| Diabetes % | 943 | 167(17.70%) | 212(22.50%) | 251(26.60%) | 313(33.20%) | <0.001 |
| Hypertension % | 1992 | 511(25.70%) | 501(25.20%) | 468(23.50%) | 512(25.70%) | 0.269 |
| Hyperlipidemia % | 3300 | 729(21.90%) | 790(23.70%) | 861(25.90%) | 950(28.50%) | <0.001 |
| Hyperuricemia % | 1011 | 183(18.10%) | 225(22.30%) | 280(27.70%) | 323(31.90%) | <0.001 |
| MetS % | | | | | | <0.001 |
| Yes | 1717 | 296(17.20%) | 366(21.30%) | 446(26.00%) | 609(35.50%) | |
| No | 3200 | 931(29.10%) | 867(27.10%) | 779(24.30%) | 623(19.50%) | |
| Elevated FPG % | | | | | | <0.001 |
| Yes | 2542 | 537(21.10%) | 612(24.10%) | 670(26.40%) | 723(28.40%) | |
| No | 2375 | 690(29.10%) | 621(26.10%) | 555(23.40%) | 509(21.40%) | |
| Elevated BP % | | | | | | 0.403 |
| Yes | 2174 | 551(25.30%) | 547(25.20%) | 517(23.80%) | 559(25.70%) | |
| No | 2743 | 676(24.60%) | 686(25.00%) | 708(25.80%) | 673(24.50%) | |
| Elevated TG % | | | | | | <0.001 |
| Yes | 1077 | 176(16.30%) | 220(20.40%) | 302(28.00%) | 379(35.20%) | |
| No | 3840 | 1051(27.40%) | 1013(26.40%) | 923(24.00%) | 853(22.20%) | |
| Low HDL-C % | | | | | | <0.001 |
| Yes | 1540 | 259(16.80%) | 333(21.60%) | 395(25.60%) | 553(35.90%) | |
| No | 3377 | 968(28.70%) | 900(26.70%) | 830(24.60%) | 679(20.10%) | |
| Elevated WC % | | | | | | <0.001 |
| Yes | 2621 | 532(20.30%) | 633(24.20%) | 676(25.80%) | 780(29.80%) | |
| No | 2296 | 695(30.30%) | 600(26.10%) | 549(23.90%) | 452(19.70%) | |

**Table 2. Association between WMR and other parameters.**

| | r | p |
|---|---|---|
| Age, years | -0.011 | 0.426 |
| Gender | 0.006 | 0.682 |
| WC, cm | 0.197** | <0.001 |
| BMI | 0.174** | <0.001 |
| WHtR | 0.211** | <0.001 |
| SBP, mm Hg | 0.008 | 0.619 |
| DBP, mm Hg | -0.024 | 0.122 |
| FPG, mg/dL | 0.147** | <0.001 |
| 2hPG | 0.153** | <0.001 |
| HbA1c, % | 0.134** | <0.001 |
| HOMA-IR | 0.220** | <0.001 |
| TC, mg/dL | 0.008 | 0.598 |
| TG, mg/dL | 0.228** | <0.001 |
| LDL-C, mg/dL | 0.018 | 0.212 |
| HDL-C, mg/dL | -0.217** | <0.001 |
| WBC, $\times 10^9$/L | 0.923** | <0.001 |
| RBC, $\times 10^{12}$/L | 0.109** | <0.001 |
| Hb, mg/dL | 0.086** | <0.001 |
| PLT, $\times 10^9$/L | 0.457** | <0.001 |
| ALT, U/L | -0.071** | <0.001 |
| AST, U/L | 0.030* | 0.037 |
| GGT, U/L | 0.113** | <0.001 |
| Uric acid,mg/dL | 0.110** | <0.001 |
| MetS | 0.199** | <0.001 |

(OR) = 4.049; 95% confidence interval (CI), 3.214–5.100; P<0.001), the least adjusted model (model 2) (OR = 4.272; 95% CI, 3.363–5.4279; P<0.001), and the fully adjusted model (model 3) (OR = 3.185; 95% CI, 2.399–4.229; P<0.001). This indicates that each standard deviation increase in WMR was associated with a significant 3.185-fold increased odds of having MetS.

To further explore this association, WMR was categorized into quartiles for sensitivity analysis. Participants in the highest quartile (Q4) had a statistically significant 2.548-fold increased risk of MetS compared to those in the lowest quartile (Q1) (OR = 2.548; 95% CI, 2.067–3.140; P<0.001). Participants in the Q2 and Q3 groups also showed elevated risks of MetS prevalence, with ORs of 1.285 (95% CI, 1.045–1.582; P<0.001) and 1.586 (95% CI, 1.288–1.953; P<0.001), respectively, compared to the Q1 group.

Additionally, Table 4 details the association between WMR and five biochemical indicators related to MetS across different models. Using multivariate regression analysis with a complex sampling design, we observed that higher WMR levels were significantly associated with elevated levels of FPG, TG, and WC, as well as reduced levels of HDL-C. For instance, the risk of elevated FPG increased by 1.571-fold (95% CI, 1.280–1.928), 1.638-fold (95% CI, 1.326–2.023), and 2.280-fold (95% CI, 1.833–2.837) in the Q2, Q3, and Q4 groups, respectively (P<0.001). Similar significant associations were found for elevated TG, Low HDL-C, and elevated WC across quartile groups, all indicating heightened risks associated with increasing WMR levels.

Table 3. Univariate analysis of determinants of MetS in study subjects.

| Variable | Univariate analysis | |
|---|---|---|
| | Statistic | P |
| Age | -15.215 | <0.001 |
| Gender | 10.447 | 0.001 |
| Race % | 66.169 | <0.001 |
| Education % | 35.156 | <0.001 |
| Smoking status % | 37.493 | <0.001 |
| Alcohol consumption % | 12.047 | 0.002 |
| Physically active % | 97.657 | <0.001 |
| PIR | -2.748 | 0.006 |
| RBC, ×1012/L | -3.576 | <0.001 |
| Hb, mg/dL | -0.652 | 0.514 |
| ALT, U/L | -1.371 | 0.170 |
| AST, U/L | -11.237 | <0.001 |
| GGT, U/L | -15.823 | <0.001 |
| TC, mg/dL | -6.789 | <0.001 |
| LDL-C, mg/dL | -6.767 | <0.001 |
| Uric acid,mg/dL | -12.799 | <0.001 |
| WMR | -13.953 | <0.001 |

## Subgroup analysis

Subgroup analysis was conducted based on gender, age, race, education level, smoking status, and physical activity (Fig 2). Our analysis revealed a consistent relationship between WMR levels and MetS (Fig 2A), with the risk of MetS increasing as WMR increased. This relationship was statistically significant across different subgroups (P < 0.05). However, the relationship between WMR levels and Elevated FPG (Fig 2B), Elevated TG (Fig 2D), Low HDL-C (Fig 2E), and Elevated WC (Fig 2F) varied among subgroups. Regardless of statistical significance, WMR levels were positively correlated with Elevated FPG, Elevated TG, Low HDL-C, and Elevated WC. In contrast, the relationship between WMR levels and Elevated BP (Fig 2C) was not statistically significant across subgroups (P > 0.05).

Interaction tests revealed that gender, age, race, education, smoking status, and physical activity might influence the positive correlation between WMR and MetS, as well as Elevated WC (interaction P < 0.05). There was no significant difference in the correlations between WMR and Elevated FPG across gender, education, and smoking status, between WMR and Elevated BP across gender, race, education, smoking status, and physical activity, and between WMR and Elevated TG across gender, and between WMR and Low HDL-C across physical activity. This indicates that these factors do not significantly modify the positive correlations (all interactions > 0.05).

## Predictive value of WMR in screening for the presence of MetS and its components

To further explore the predictive value of WMR for MetS and its components, ROC curve analysis was performed. As shown in Fig 3, the optimal cut-off value of WMR for predicting the presence of MetS was 0.7974 (sensitivity: 58.4%; specificity: 59.9%; AUC: 0.621; Fig 3A). The optimal cut-off value of WMR for predicting elevated FPG was 0.8009 (sensitivity: 51.2%; specificity: 60.0%; AUC: 0.570; Fig 3B). WMR did not have statistically significant value in predicting the presence of elevated BP. The optimal cut-off value of WMR for predicting elevated TG was 0.7798 (sensitivity: 63.1%; specificity: 54.0%; AUC: 0.612; Fig 3D). The optimal cut-off value of WMR for predicting low HDL-C was 0.8475

**Table 4. Association of WMR with Metabolic syndrome (MetS) and its components.**

| | Model1 | | Model2 | | Model3 | |
|---|---|---|---|---|---|---|
| | OR (95%CI) | P value | OR (95%CI) | P value | OR (95%CI) | P value |
| MetS | | | | | | |
| Continous | 4.049(3.214,5.100) | <0.001 | 4.272(3.363,5.427) | <0.001 | 3.185(2.399,4.229) | <0.001 |
| Q1 | Ref. | | Ref. | | Ref. | |
| Q2 | 1.328(1.110,1.588) | 0.002 | 1.325(1.101,1.593) | 0.003 | 1.285(1.045,1.582) | <0.001 |
| Q3 | 1.801(1.512,2.145) | <0.001 | 1.781(1.485,2.136) | <0.001 | 1.586(1.288,1.953) | <0.001 |
| Q4 | 3.075(2.589,3.652) | <0.001 | 3.204(2.679,3.832) | <0.001 | 2.548(2.067,3.140) | <0.001 |
| P for trend | <0.001 | | <0.001 | | <0.001 | |
| Elevated FPG | | | | | | |
| Continous | 2.343(1.877,2.924) | <0.001 | 2.833(2.218,3.618) | <0.001 | 2.126(1.599,2.826) | <0.001 |
| Q1 | Ref. | | Ref. | | Ref. | |
| Q2 | 1.266(1.080,1.484) | 0.004 | 1.288(1.081,1.535) | 0.005 | 1.571(1.280,1.928) | <0.001 |
| Q3 | 1.551(1.323,1.819) | <0.001 | 1.611(1.350,1.922) | <0.001 | 1.638(1.326,2.023) | <0.001 |
| Q4 | 1.825(1.555,2.142) | <0.001 | 2.079(1.741,2.483) | <0.001 | 2.280(1.833,2.837) | <0.001 |
| P for trend | <0.001 | | <0.001 | | <0.001 | |
| Elevated BP | | | | | | |
| Continous | 0.977(0.806,1.186) | 0.054 | 0.997(0.821,1.210) | 0.973 | 0.988(0.785,1.245) | 0.921 |
| Q1 | Ref. | | Ref. | | Ref. | |
| Q2 | 0.978(0.834,1.147) | 0.787 | 0.990(0.843,1.162) | 0.901 | 0.969(0.812,1.157) | 0.728 |
| Q3 | 0.896(0.764,1.051) | 0.177 | 0.914(0.777,1.074) | 0.274 | 0.925(0.772,1.110) | 0.402 |
| Q4 | 1.019(0.869,1.194) | 0.816 | 1.047(0.892,1.230) | 0.572 | 1.008(0.837,1.214) | 0.933 |
| P for trend | 0.907 | | 0.807 | | 0.942 | |
| Elevated TG | | | | | | |
| Continous | 3.009(2.352,3.848) | <0.001 | 2.831(2.197,3.649) | <0.001 | 2.893(2.095,3.995) | <0.001 |
| Q1 | Ref. | | Ref. | | Ref. | |
| Q2 | 1.297(1.045,1.610) | <0.001 | 1.179(0.946,1.469) | <0.001 | 1.260(0.959,1.655) | 0.098 |
| Q3 | 1.954(1.590,2.400) | <0.001 | 1.730(1.402,2.135) | <0.001 | 1.764(1.354,2.298) | <0.001 |
| Q4 | 2.653(2.172,3.242) | <0.001 | 2.436(1.984,2.990) | <0.001 | 2.548(1.952,3.325) | <0.001 |
| P for trend | <0.001 | | <0.001 | | <0.001 | |
| Low HDL-C | | | | | | |
| Continous | 3.977(3.149,5.022) | <0.001 | 3.768(2.975,4.772) | <0.001 | 2.770(2.049,3.744) | <0.001 |
| Q1 | Ref. | | Ref. | | Ref. | |
| Q2 | 1.383(1.148,1.666) | 0.001 | 1.345(1.115,1.624) | 0.002 | 1.293(1.031,1.620) | 0.026 |
| Q3 | 1.779(1.483,2.134) | <0.001 | 1.688(1.403,2.031) | <0.001 | 1.444(1.148,1.815) | 0.002 |
| Q4 | 3.044(2.550,3.634) | <0.001 | 2.920(2.439,3.495) | <0.001 | 2.249(1.794,2.820) | <0.001 |
| P for trend | <0.001 | | <0.001 | | <0.001 | |
| Elevated WC | | | | | | |
| Continous | 2.914(2.321,3.659) | <0.001 | 3.674(2.850,4.736) | <0.001 | 2.678(1.969,3.643) | <0.001 |
| Q1 | Ref. | | Ref. | | Ref. | |
| Q2 | 1.378(1.176,1.615) | <0.001 | 1.597(1.338,1.906) | <0.001 | 1.571(1.280,1.928) | <0.001 |
| Q3 | 1.609(1.371,1.887) | <0.001 | 1.876(1.569,2.244) | <0.001 | 1.638(1.326,2.023) | <0.001 |
| Q4 | 2.254(1.918,2.650) | <0.001 | 2.718(2.268,3.258) | <0.001 | 2.280(1.833,2.837) | <0.001 |
| P for trend | <0.001 | | <0.001 | | <0.001 | |

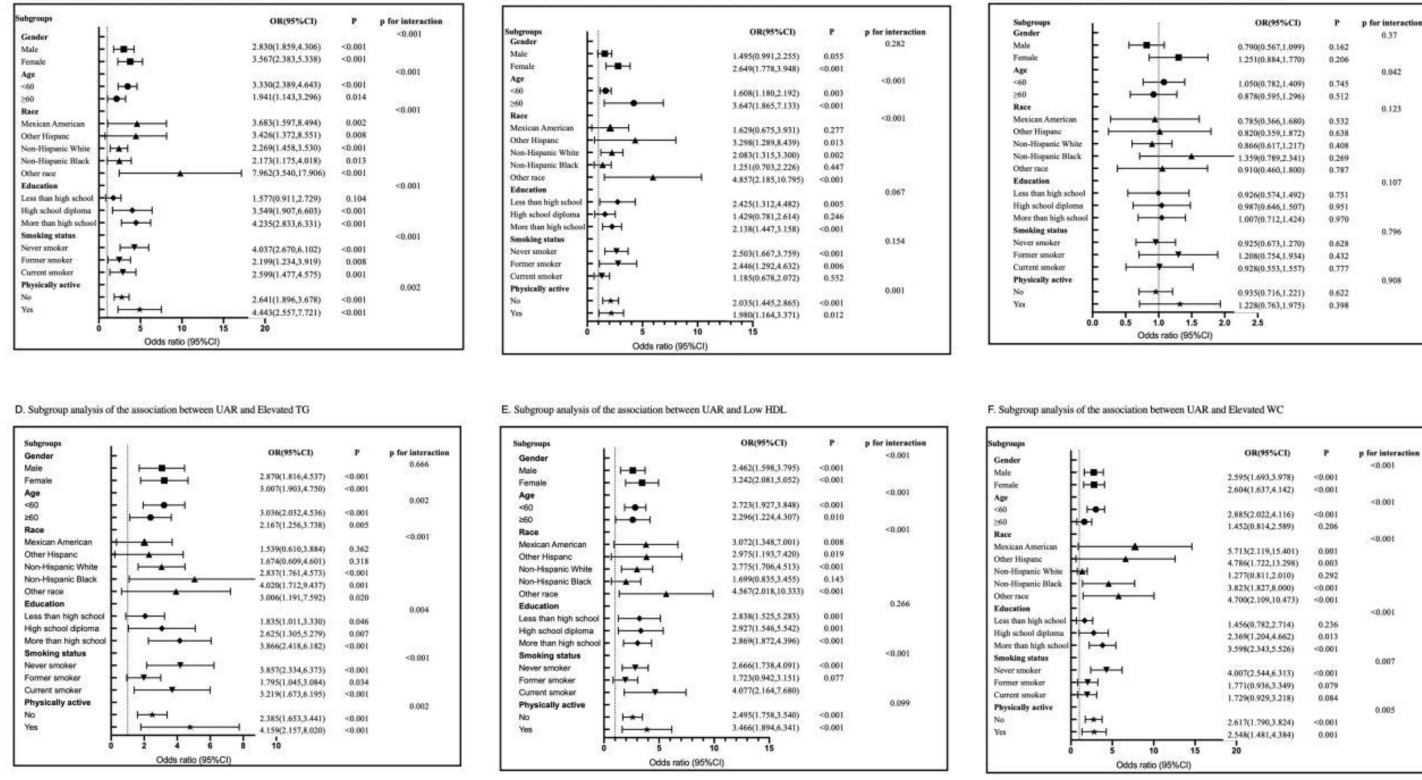

**Fig 2. Subgroup analysis of the association between WMR and MetS and its components.**

(sensitivity: 49.7%; specificity: 67.5%; AUC: 0.617; Fig 3E). The optimal cut-off value of WMR for predicting elevated WC was 0.741 (sensitivity: 63.3%; specificity: 49.7%; AUC: 0.586; Fig 3F).

## Discussion

This study investigated the correlation between MetS and WMR. The main finding is that WMR levels are significantly higher in patients with MetS. Moreover, WMR levels are positively correlated with MetS and its components, including elevated FPG, elevated TG, low HDL-C, and elevated WC. This correlation remains significant even after controlling for confounding variables, and is consistently observed in both continuous and categorical analyses. Our findings provide new insights and robust evidence for further clinical and basic research.

In recent years, there has been increasing attention on the significance of common indicators in routine blood tests for diagnosing and preventing metabolic diseases. Inflammation and oxidative stress are pivotal in the pathogenesis of metabolic complications such as hyperlipidemia, hypertension, and impaired glucose tolerance, all contributing to metabolic dysfunction [15]. Yang Zhao et al. previously explored the correlation between the Systemic Immune-Inflammation Index (SII) and MetS, highlighting a positive association between SII and MetS risk [16]. WMR (derived from WBC and MPV) has emerged as a novel inflammatory marker in studies. Our study identified WMR as significantly positively correlated with MetS. Unlike findings by Yang Zhao et al., who observed strong positive correlations between SII and abdominal obesity, hypertension, and a negative correlation with HDL-C, SII scores did not significantly correlate with fasting blood glucose and serum TG. Conversely, our study revealed significant positive correlations between WMR levels and fasting

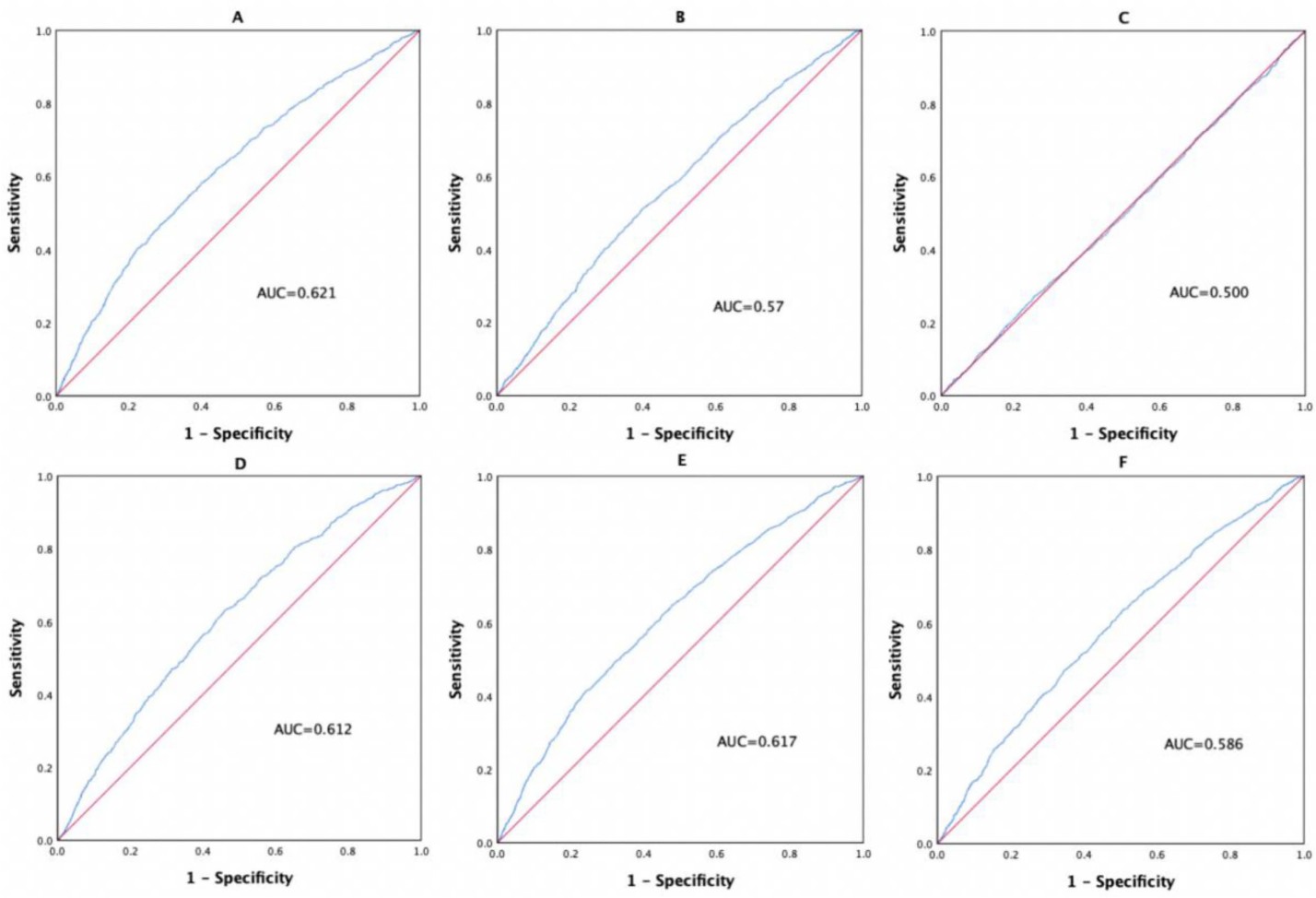

**Fig 3. ROC curve analysis of WMR for predicting MetS and its components.**

blood glucose, abdominal obesity, and TG, alongside a negative correlation with HDL-C, but not hypertension. These similarities and discrepancies lead us to speculate on potential complementary roles of these two indicators.

Existing studies have not reported correlations between WMR and blood glucose, TG, HDL-C, or WC. This study found that WMR is significantly positively correlated with FPG, TG, WC, and negatively correlated with HDL-C. The exact mechanisms are currently unclear. After reviewing the literature, it is speculated to be related to the following mechanisms: The relationship between hyperglycemia and chronic inflammation has been reported in multiple animal and clinical trials, where increased blood glucose can promote inflammation [16,17]. Previous studies have explored systemic inflammation and hyperlipidemia, such as Nayili Mahemuti's cross-sectional study which showed that SII is associated with a higher risk of hyperlipidemia [18]. Regarding the relationship between inflammatory response and lipid levels, some previous studies have elucidated the association between inflammation and lipid levels [19]. Ma et al. reported that elevated plasma C-reactive protein (CRP) levels were associated with higher TG and lower HDL-C concentrations [20]. Activation of the inflammatory cascade induces a decrease in HDL-C, impaired reverse cholesterol transport, as well as parallel changes in lipoproteins, enzymes, antioxidant capacity, and ATP-binding cassette A1-dependent efflux. This reduction in HDL-C and phospholipids may stimulate compensatory changes, as the synthesis and accumulation of phospholipid-rich VLDL

can combine with bacterial products and other toxins, leading to hypertriglyceridemia [21]. Multiple preclinical and clinical studies have confirmed that chronic low-grade inflammation in adipose tissue is mechanistically linked to metabolic disorders and organ complications in overweight and obese individuals [22]. Obesity is associated with increased white adipose tissue, which is the primary fat storage depot and the largest endocrine organ secreting adipokines and cytokines throughout the body. Weight gain and obesity lead to a phenotypic transformation of white adipose tissue, characterized by the appearance of inflamed and dysfunctional adipocytes and infiltration of immune cells in the stromal vascular fraction [23,24]. Inflamed adipocytes locally and systemically secrete pro-inflammatory cytokines, which in turn disrupt the normal function of adipose tissue itself and distant organs. Furthermore, even in seemingly healthy individuals with normal weight, inflamed white adipose tissue can cause widespread systemic inflammation through cytokine release [22].

In addition to this, we found that the association between WMR and Elevated BP was not significant, even after adjusting for covariates. Although some studies suggest an association between hypertension and chronic inflammation [16], it is speculated that changes in patients' unhealthy lifestyle habits and medication use following the diagnosis of hypertension may have influenced the observed results.

In conclusion, chronic inflammation is closely associated with MetS and its components, making the improvement of inflammation a potential method for preventing and treating MetS. The novel inflammatory marker WMR holds promise as a biomarker for metabolic syndrome; however, its clinical value and applications need further confirmation through additional basic and clinical research.

Our study has several notable strengths. Firstly, it is the first to investigate the relationship between WMR and MetS. Secondly, we analyzed the associations between WMR and each component of MetS, providing detailed explanations of our findings. This approach demonstrates the rigor of our study. Additionally, we utilized a large, nationally representative cross-sectional survey, enhancing the representativeness of our results.

However, it should be noted that our study still has several limitations. Firstly, it is cross-sectional in nature, which means that we cannot establish a causal relationship between WMR and MetS. Secondly, the underlying mechanism linking WMR to MetS requires further investigation through prospective large-scale studies. Despite these limitations, the relatively large sample size enhances the robustness of our findings. Given that WMR can be easily calculated from routine indicators, it holds potential for practical use in clinical settings, especially in large-scale screening procedures.

## Conclusions

Increasing WMR levels are significantly associated with the risk of MetS and its components: Elevated FPG, Elevated TG, Low HDL-C, and Elevated WC. This suggests that WMR could potentially serve as a valuable and reliable biomarker for MetS, highlighting the importance of closely monitoring patients with elevated WMR to improve prevention and mitigate the development of MetS. However, prospective cohort studies are warranted to confirm these associations and to further explore the causal relationships between WMR and the development of MetS.

## Author contributions

Shuiying Huang analyzed the data and performed the data curation, and drafted the manuscript. Yi Wang analyzed the data and data curation. Yu Wang and Ying Miao drafted the manuscript, Oversaw the study design and revised the paper.

## Author contributions

**Conceptualization:** Shuiying Huang, Yu Wang.

**Methodology:** Shuiying Huang, Yi Wang, Yu Wang, Ying Miao.

**Validation:** Yi Wang.

**Visualization:** Yu Wang, Ying Miao.

**Writing – original draft:** Shuiying Huang, Yu Wang.

**Writing – review & editing:** Yu Wang.

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
