## [Decision Letter · Decision Letter 0]

4 Jul 2025

Dear Dr. Wang,

The manuscript requires minor but CRITICAL revisionClarity in the theoretical and biological/physiological backing of the results is very importantThe introduction and discussion sections in particular should be enriched to create a logical and biological justification of the results.

plosone@plos.org. . . . A rebuttal letter that responds to each point raised by the academic editor and reviewer(s). You should upload this letter as a separate file labeled 'Response to Reviewers'.A marked-up copy of your manuscript that highlights changes made to the original version. You should upload this as a separate file labeled 'Revised Manuscript with Track Changes'.An unmarked version of your revised paper without tracked changes. You should upload this as a separate file labeled 'Manuscript'.

We look forward to receiving your revised manuscript.

Kind regards,

Fredirick Lazaro mashili, MD, PhD

Academic Editor

PLOS ONE

Journal Requirements:

NO authors have competing interests

4. Thank you for uploading your study's underlying data set. Unfortunately, the repository you have noted in your Data Availability statement does not qualify as an acceptable data repository according to PLOS's standards.

At this time, please upload the minimal data set necessary to replicate your study's findings to a stable, public repository (such as figshare or Dryad) and provide us with the relevant URLs, DOIs, or accession numbers that may be used to access these data. For a list of recommended repositories and additional information on PLOS standards for data deposition, please see https://journals.plos.org/plosone/s/recommended-repositories....

5. Please upload a new copy of Figure 2 as the detail is not clear. Please follow the link for more information: ""https://blogs.plos.org/plos/2019/06/looking-good-tips-for-creating-your-plos-figures-graphics/"" https://blogs.plos.org/plos/2019/06/looking-good-tips-for-creating-your-plos-figures-graphics/

Additional Editor Comments:

Please address thoroughtly all the comments and concerns from the reviewers. The study is very well designed and conducted, and the manuscript well written. The main issue is complexity of data and lack of clarity and logic behind the theoretical framework that back the results.

Reviewers' comments:

Reviewer's Responses to Questions

**Comments to the Author**

1. Is the manuscript technically sound, and do the data support the conclusions?

Reviewer #1: Yes

Reviewer #2: Yes

2. Has the statistical analysis been performed appropriately and rigorously?

Reviewer #1: Yes

Reviewer #2: Yes

3. Have the authors made all data underlying the findings in their manuscript fully available?

Reviewer #1: Yes

Reviewer #2: Yes

4. Is the manuscript presented in an intelligible fashion and written in standard English?

Reviewer #1: Yes

Reviewer #2: Yes

Reviewer #1: Dear Authors,

I have reviewed the manuscript titled “Association Between the Novel Inflammatory Marker White Blood Cell Count-to-Mean Platelet Volume Ratio and Metabolic Syndrome: A Population-Based Study from NHANES 2011–2020” which investigates association between WMR and the risk of developing Metabolic syndrome.

General Comments

Overall, the manuscript is well-written, and I appreciate the thoroughness with which you have conducted your research. There are several aspects that would benefit from clarification or revision to strengthen the quality of the paper. Below, I provide both comments on the manuscript.

1. Title and Abstract

• Title:

o The title does not clearly indicate that the study is cross-sectional.

• Abstract:

o The abstract provides a concise summary of the study, including the background, objectives, methods, key findings, and conclusions. However, it would benefit from a clearer mention of the study design (cross-sectional). Besides, I suggest revising the conclusion to better emphasize on performing of prospective studies in the conclusion section.

2. Introduction

• Introduction: the background is generally well-explained.

• First paragraph: The same references (first) should not be used for consecutive sentences; the first reference should be removed.

3. Methodology

• The study design is appropriate for the research objectives. However, the exclusion criteria regarding WBC could be more clearly explained. Specifically, I recommend elaborating on the exclusion criteria of the critical influencing factors, including chronic inflammatory disease, cancer, drugs, and others that could affect WBC measurement accuracy.

• Number of participants is different from flowchart and abstract.

The definition of cigarette smoking could be clarified.

4. Results

• The presentation of the results is clear.

5. Discussion and Conclusion

• In the discussion is comprehensive.

Overall Evaluation

In conclusion, I believe the manuscript provides valuable contributions to the field. The key findings are promising, but the clarity of certain sections and some methodological details require refinement to improve the manuscript's overall rigor.

Yours sincerely,

Reviewer #2: The study was well designed, executed, data robustly crunched and analyzed. The manuscript is generally very well written. However, being mostly an association study, that involve a mathematically modeled marker (not a direct biological marker) more inputs and clarifications will strengthen the manuscript and boost the quality of this evidence.

The gap is mainly lack of properly synthesized justification that is based on biological plausibility and previous evidence. Being one of the earlier studies to bring forward these results, a well written justification in the introduction and discussions is important. The authors should describe known biological facts, established evidence in vitro and in vivo, and synthesize a logical framework in the introduction section. This should include related literature from related fields if no direct literature is available. The discussion section should also feature what has been introduced and discussing it based on the current results. Results can be confusing, and at times draw people out of focus if not properly justified. While these results are likely novel, with potential to drive a paradigm shift towards important discoveries, they can also be confusing and cause a negative paradigm shift. The authors should read, understand and present well the theoretical support to their results.

.

Reviewer #1: No

Reviewer #2: **Yes:**Fredirick mashiliFredirick mashiliFredirick mashiliFredirick mashili

While revising your submission, please upload your figure files to the Preflight Analysis and Conversion Engine (PACE) digital diagnostic tool, https://pacev2.apexcovantage.com/. PACE helps ensure that figures meet PLOS requirements. To use PACE, you must first register as a user. Registration is free. Then, login and navigate to the UPLOAD tab, where you will find detailed instructions on how to use the tool. If you encounter any issues or have any questions when using PACE, please email PLOS at . PACE helps ensure that figures meet PLOS requirements. To use PACE, you must first register as a user. Registration is free. Then, login and navigate to the UPLOAD tab, where you will find detailed instructions on how to use the tool. If you encounter any issues or have any questions when using PACE, please email PLOS at . PACE helps ensure that figures meet PLOS requirements. To use PACE, you must first register as a user. Registration is free. Then, login and navigate to the UPLOAD tab, where you will find detailed instructions on how to use the tool. If you encounter any issues or have any questions when using PACE, please email PLOS at . PACE helps ensure that figures meet PLOS requirements. To use PACE, you must first register as a user. Registration is free. Then, login and navigate to the UPLOAD tab, where you will find detailed instructions on how to use the tool. If you encounter any issues or have any questions when using PACE, please email PLOS at figures@plos.org. Please note that Supporting Information files do not need this step.. Please note that Supporting Information files do not need this step.

---

## [Author Response · Author response to Decision Letter 1]

15 Jul 2025

Dear Editor,

Thank you for taking the time to review my manuscript amidst your busy schedule. I have carefully addressed the comments you provided and made the corresponding revisions. The detailed revisions are as follows:

1.Please ensure that your manuscript meets PLOS ONE's style requirements, including those for file naming.

In the revised manuscript, we have modified the article according to PLOS ONE's style requirements.

2.Thank you for stating the following in the Competing Interests section: NO authors have competing interests

We have included this explanation in the revised manuscript.

3.Your ethics statement should only appear in the Methods section of your manuscript. If your ethics statement is written in any section besides the Methods, please move it to the Methods section and delete it from any other section. Please ensure that your ethics statement is included in your manuscript, as the ethics statement entered into the online submission form will not be published alongside your manuscript.

In accordance with the requirements, we have relocated the ethics statement to the Methods section in the revised manuscript.

4. Thank you for uploading your study's underlying data set. Unfortunately, the repository you have noted in your Data Availability statement does not qualify as an acceptable data repository according to PLOS's standards.

We have uploaded the data to Figshare in accordance with the requirements. Kindly verify whether it meets the journal’s standards.

URL：https://figshare.com/account/articles/29557250?file=56236955

DOI ：10.6084/m9.figshare.29557250

5. Please upload a new copy of Figure 2 as the detail is not clear. Please follow the link for more information: "https://blogs.plos.org/plos/2019/06/looking-good-tips-for-creating-your-plos-figures-graphics/"

We have re-uploaded Figure 2.

We have thoroughly reviewed the reference list and confirmed that it contains no retracted publications.

Dear Reviewers,

I sincerely appreciate your thorough and thoughtful review of my manuscript and the valuable suggestions you have provided. Your insightful feedback has been instrumental in enhancing the quality of my manuscript. I have now revised the manuscript in accordance with your recommendations. The detailed revisions are as follows:

1. • Title: The title does not clearly indicate that the study is cross-sectional.

• Abstract: The abstract provides a concise summary of the study, including the background, objectives, methods, key findings, and conclusions. However, it would benefit from a clearer mention of the study design (cross-sectional). Besides, I suggest revising the conclusion to better emphasize on performing of prospective studies in the conclusion section.

•In the revised manuscript, we modified the title to clearly indicate that this is a cross-sectional study.

•In the revised abstract, we added in the Methods section that this is a cross-sectional study. Furthermore, in the conclusion section, we emphasized the need for conducting prospective studies.

2.Introduction：First paragraph: The same references (first) should not be used for consecutive sentences; the first reference should be removed.

In the revised manuscript, we have removed the first reference.

3. Methodology

• The study design is appropriate for the research objectives. However, the exclusion criteria regarding WBC could be more clearly explained. Specifically, I recommend elaborating on the exclusion criteria of the critical influencing factors, including chronic inflammatory disease, cancer, drugs, and others that could affect WBC measurement accuracy.

• Number of participants is different from flowchart and abstract.

• The definition of cigarette smoking could be clarified.

•In the revised manuscript, we added relevant explanations in the Data Collection section of the Methods.

•We appreciate your careful observation, and have corrected the number of participants in the Methods section to align with the flowchart and abstract.

•In the revised manuscript, the Methods section includes an added description of smoking classification.

---

## [Decision Letter · Decision Letter 1]

15 Feb 2026

Association between the novel inflammatory marker white blood cell count-to-mean platelet volume ratio and metabolic syndrome: a cross-sectional study based on NHANES 2011–2020

PONE-D-24-46411R1

Dear Dr. Wang,

We’re pleased to inform you that your manuscript has been judged scientifically suitable for publication and will be formally accepted for publication once it meets all outstanding technical requirements.

Kind regards,

Fredirick Lazaro mashili, MD, PhD

Academic Editor

PLOS One

Additional Editor Comments (optional):

All the reviewers comments have been sufficiently addressed.

Reviewers' comments:

Reviewer's Responses to Questions

**Comments to the Author**

Reviewer #1: All comments have been addressed

Reviewer #2: All comments have been addressed

2. Is the manuscript technically sound, and do the data support the conclusions?

Reviewer #1: Yes

Reviewer #2: Yes

3. Has the statistical analysis been performed appropriately and rigorously?

Reviewer #1: Yes

Reviewer #2: Yes

4. Have the authors made all data underlying the findings in their manuscript fully available?

Reviewer #1: Yes

Reviewer #2: Yes

5. Is the manuscript presented in an intelligible fashion and written in standard English?

Reviewer #1: Yes

Reviewer #2: (No Response)

Reviewer #1: Dear Authors,

Thank you for your comprehensive responses to my comments. I appreciate the effort you put into addressing the suggestions and improving the manuscript. I am pleased to support its acceptance.

Sincerely,

Reviewer #2: All previously raised concerns have been adequately and thoughtfully addressed. The authors have substantially strengthened both the presentation and the scientific rigor of the manuscript. The revised version is clearer, more coherent, and methodologically sound.

.

Reviewer #1: No

Reviewer #2: **Yes:**Fredirick MashiliFredirick MashiliFredirick MashiliFredirick Mashili

---

## [Editor Report · Acceptance letter]

PONE-D-24-46411R1

PLOS One

Dear Dr. Wang,

I'm pleased to inform you that your manuscript has been deemed suitable for publication in PLOS One. Congratulations! Your manuscript is now being handed over to our production team.

Kind regards,

on behalf of

Dr. Fredirick Lazaro mashili

Academic Editor

PLOS One